# Modification and Solubility Enhancement of Rice Protein and Its Application in Food Processing: A Review

**DOI:** 10.3390/molecules28104078

**Published:** 2023-05-13

**Authors:** Jingjing Yang, Dan Meng, Zijian Wu, Jinyu Chen, Lu Xue

**Affiliations:** Tianjin Key Laboratory of Food Biotechnology, College of Biotechnology and Food Science, Tianjin University of Commerce, Tianjin 300134, China; 120200310@stu.tjcu.edu.cn (J.Y.);

**Keywords:** rice protein, solubility enhancement, food processing, application

## Abstract

Rice protein is a high-quality plant-based protein source that is gluten-free, with high biological value and low allergenicity. However, the low solubility of rice protein not only affects its functional properties such as emulsification, gelling, and water-holding capacity but also greatly limits its applications in the food industry. Therefore, it is crucial to modify and improve the solubility of rice protein. In summary, this article discusses the underlying causes of the low solubility of rice protein, including the presence of high contents of hydrophobic amino acid residues, disulfide bonds, and intermolecular hydrogen bonds. Additionally, it covers the shortcomings of traditional modification methods and the latest compound improvement methods, compares various modification methods, and puts forward the best sustainable, economical, and environmentally friendly method. Finally, this article lists the uses of modified rice protein in dairy, meat, and baked goods, providing a reference for the extensive application of rice protein in the food industry.

## 1. Introduction

Rice protein is a high-quality, gluten-free plant-based protein derived from byproducts of rice processing. It contains a comprehensive range of amino acids (including all 20 essential and non-essential amino acids), a well-balanced ratio, and one of the highest biological values of all cereal crops [1]. Additionally, rice protein is characterized by its light color, mild odor, and low allergenicity [2]. Moreover, rice protein hydrolysates have been reported to possess various biological activities, such as antihypertensive, antioxidant, anticancer, and anti-obesity effects [3,4,5,6]. However, the poor water solubility of rice protein significantly restricts its applications in the food industry and related fields.

Protein solubility typically refers to the concentration of protein in a saturated aqueous solution in equilibrium with the solid phase (crystalline or amorphous) under external conditions such as pH, temperature, and ionic strength [7]. Water solubility is one of the important thermodynamic properties of proteins, and it is a prerequisite for other functional properties of proteins, such as emulsification, foaming, gelation, and digestibility. It also further determines the texture and stability of processed food products. However, rice protein has a relatively low overall water solubility, and the high content of alkaline-soluble glutelin makes it difficult to dissolve in slightly acidic or neutral solutions, resulting in low utilization efficiency [8]. This greatly affects its applications in the food and related fields. In order to improve the solubility and utilization of rice protein, various physical (ultrasound, high-pressure, and microwave), chemical (glycosylation, phosphorylation, and deamidation), and biological enzyme-based modification methods can be used, and these techniques can also be combined to produce a simple, safe, and efficient modification of rice protein. The modified rice protein thus obtained has great potential for application in food processing [9].

This article provides an overview of recent methods used for modifying and increasing the solubility of rice protein and further explores the application of modified rice protein in food production, such as dairy products, meat products, and baked goods.

## 2. Analysis of the Water Solubility Characteristics and Reasons for Rice Protein

According to the solubility of each constituent protein, rice protein can be divided into four categories, of which the most abundant (about 80%) is alkali-soluble globulin, which is mainly composed of 30–40 kDa α (acidic) subunit and 17–23 kDa β (basic) subunit. The other categories include 1–5% alcohol-soluble protein, 2–10% salt-soluble globulin, and 2–5% water-soluble albumin. The main subunits are 11~24 kDa, 19–25 kDa, and 10~100 kDa, respectively [9,10]. The properties of the predominant alkali-soluble glutelin determine the overall properties of rice protein.

The low water solubility of rice glutelin is primarily due to the presence of many hydrogen bonds between and within its molecules, which are formed by the amide groups on the glutamine and asparagine residues interacting with the smaller-sized glycine and alanine residues through steric hindrance [8]. Figure 1 shows the primary structure and amino acid residues composition of rice glutelin from three rice sources (*Oryza sativa*, *Oryza sativa* Japonica Group, and *Oryza sativa* Indica Group). It can be found that rice glutelin contains 17.23–18.60% of glutamine and asparagine residues, and the hydrogen bonds formed by these two amino acid residues replace a large number of hydrogen bonds between amide groups and water molecules, thereby reducing the water solubility of rice glutelin [11].

Secondly, there are disulfide bonds between rice glutelin subunits, which are formed by the coupling of thiol groups of two different cysteine residues. As shown in Figure 1, rice glutelin contains a small number of cysteine residues (1.40~2.00%), which can interact with each other to generate disulfide bonds within or between subunits, especially between subunits, leading to the formation of large molecular complexes and a decrease in the water solubility of rice glutelin [12].

In addition, rice glutelin contains 37.27~39.60% hydrophobic amino acid residues (as shown in Figure 1), which can cause hydrophobic interactions between amino acid residues and result in the formation of a stable, tensile protein network structure between subunits, which limits the entry of water molecules [13] and therefore reduces its water solubility.

## 3. Methods for Modifying Rice Protein to Increase Solubility

By taking corresponding measures to address the low water solubility of rice glutelin mentioned above, it is possible to effectively improve the water solubility of rice glutelin and thus enhance the overall solubility of rice protein. The commonly used methods include the following: (1) utilizing the interaction of polysaccharides, polyphenols, composite proteins, and phosphates with rice protein to increase the number of hydrophilic hydroxyl groups or hydrogen bonds formed with water molecules. (2) Using deamidation reactions to break the amide bonds in the side chains of rice protein and reduce the formation of hydrogen bonds between protein subunits. (3) Enzymatically hydrolyzing rice protein to break down the large protein molecules into small peptide molecules and promote their binding to water molecules.

### 3.1. Modification and Solubility Enhancement of Rice Protein by Polysaccharides

The hydroxyl groups within polysaccharide molecules have strong hydrophilic properties [14]. It is possible to effectively increase the hydrophilic properties of the protein and thus improve its water solubility through covalent or non-covalent interactions between polysaccharides and proteins [15]. For example, the Maillard reaction, in which the carbonyl group of the polysaccharide reacts with the amino group of the protein to form a condensed product [16], can create covalent bonds between protein and polysaccharide molecules. However, the compact globular structure of rice protein makes it challenging to complete the Maillard reaction with polysaccharides in a short period of time [17], thereby reducing the rate and extent of the Maillard reaction and affecting the solubility properties of the conjugated products formed by the traditional Maillard reaction [18]. In recent years, various assisting methods including ultrasound, ultra-high pressure, and microwave have been used to improve or promote the efficiency of the Maillard reaction. These techniques can successfully bring rice protein into adequate contact with polysaccharides, enhancing the reaction efficiency and water solubility. The reaction process is illustrated in Figure 2.

Ultrasound can effectively promote the mixing of protein and polysaccharide molecules, accelerate the penetration speed of solvents into proteins, and thereby increase the Maillard reaction rate [19]. Chen et al. [20] found that after 22 min of ultrasound treatment with a power of 600 W, the spatial conformation of rice protein molecules changed from a globular or blocky structure to a more uniform and loose lamellar structure, which was conducive to the entry of solvents and accelerated the Maillard reaction rate. The solubility of the modified rice protein produced by the process was significantly improved to 90.6%.

Ultra-high-pressure assistance can prevent protein denaturation and aggregation caused by long-term heating, which is more conducive to the effective entry of solvents into protein molecules during the Maillard reaction. Therefore, it is possible to improve the efficiency of the Maillard reaction at lower temperatures [21]. In addition, ultra-high pressure can cause rice protein to expand into a loose spatial conformation, promote the exposure of hydrophobic groups, increase the content of free sulfhydryl groups, and thereby enhance the gel strength of the protein [14]. Xiao et al. [22] prepared rice protein-xylose copolymers with a high grafting degree under ultra-high hydrostatic pressure assistance, which improved the solubility of rice protein (increased from 12.8% to 35.3%) and reduced the formation of harmful by-products such as acrylamide, furan, and hydroxymethyl furfural.

Proper microwave treatment can also shorten the Maillard reaction time and enhance the solubility of rice protein [23]. Meng et al. [24] found that microwave-assisted heating-assisted protein–polysaccharide binding could increase the solubility of rice bran protein to 90.97%. This is because microwaves have the ability to produce strong mechanical collisions and shearing between protein and polysaccharide molecules, which facilitated their adequate interaction. Therefore, more branched glucan may be grafted onto the protein molecules, enhancing the protein’s solubility [25].

### 3.2. Modification and Solubility Enhancement of Rice Protein by Polyphenols

Polyphenols are a group of naturally occurring compounds derived from plants that contain numerous polar hydroxyl groups, including phenolic acids, flavonoids, and tannins, among others [26]. Polyphenols can form non-covalent bonds with proteins, in which the hydrophilic phenolic hydroxyl groups in polyphenols can undergo hydrogen bonding with proteins, and the aromatic rings can interact with the hydrophobic groups of proteins through hydrophobic interactions. Polyphenols can also undergo covalent binding with amino acid residues on proteins, especially lysine residues, cysteine residues, and tryptophan residues, under alkaline conditions, thereby altering the functional properties of the protein [27].

The non-covalent binding between polyphenols and proteins can be simulated by computer docking. Therefore, in this study, AutoDock Vina docking software was used to analyze the interactions between four polyphenols (namely, gallic acid, procyanidins, resveratrol, and ferulic acid), and specific amino acid residues located at the active site of rice glutelin (gene: GLUA2), using its three-dimensional structure as a model (as shown in Figure 3). During the molecular docking process, the polyphenol is considered to be a flexible structure, while the protein is viewed as a rigid structure. The best docking result is chosen based on the principle of minimum docking energy, and Discovery Studio 2021 Client software is used for visualization display. The results showed that the four types of polyphenols mainly formed hydrogen bonds with Thr381, Arg404, Gln110, Gln160, and Gln382, and formed hydrophobic interactions with Val180, Val187, and Arg403. Among them, rice glutelin had more interaction sites with gallic acid, including Asp322, Thr381, Arg404, Gln160, and Gln386, mainly through hydrogen bonding, which was consistent with the research results of Dai [28]. The interaction between rice glutelin and ferulic acid is mainly through hydrophobic interactions, including the interaction with hydrophobic amino acid residues such as Leu407, Ile109, Ile185, Val187, Tyr379, Thr387, Phe430, and Ile428. The interaction between rice glutelin and ferulic acid can lead to a conformational change of rice glutelin, decrease the surface hydrophobicity, and increase water solubility and emulsifying properties [29].

Alkaline treatment, enzymatic treatment, radical coupling, and ultrasound-assisted methods can further improve the covalent binding efficiency between polyphenols and proteins. According to the alkaline coupling approach, phenolic hydroxyl groups are converted to quinones or semiquinones under alkaline conditions, which can further bind to more amino acid residues and alter the physicochemical properties of the target protein [30]. Wang et al. [29] found that under the condition of pH 9.0, the covalent complex of rice bran protein hydrolysate (10 mg/mL) and ferulic acid (1.5 mg/mL) had strong emulsifying (35.10%) and antioxidant properties (the DPPH clearance rate is 49.70% and the ABTS+ clearance rate is 89.04%). In addition, protein–polyphenol covalent complexes can also be produced via enzymatic and radical coupling methods [31]. However, traditional methods require long reaction times, which limits large-scale production. Ultrasound-assisted methods can improve the reaction efficiency of enzymatic, alkaline, and radical coupling methods. The underlying principle is that ultrasound treatment can cause protein spatial structure to unfold through mechanical shear, cavitation effects, and thermal effects, increasing the chance of contact reaction between protein and polyphenol molecules, and thus improving covalent reaction efficiency [32]. For example, Xue et al. [33] used ultrasound-assisted radical coupling to improve the grafting efficiency of glutelin and polyphenols, shortened the reaction time, and increased the solubility of glutelin.

### 3.3. Modification and Solubility Enhancement of Rice Protein by Interaction with Heterologous Proteins

The solubility of rice protein can also be improved by forming a complex with heterologous proteins (usually hydrophilic proteins) through pH cycling. Figure 4 depicts the mechanism of the complex formation using whey protein as an example: first, rice protein and whey protein are mixed at pH 7.0, and then the pH of the solution is adjusted to 12.0 to fully dissolve rice protein, at which point its tertiary structure is completely unfolded and can fully contact with whey protein. The protein solution is then adjusted back to pH 7.0 for protein refolding, and subsequently subjected to centrifugation, dialysis, and freeze-drying to obtain the “rice protein–whey protein” complex [34]. When rice protein and heterologous protein interact to generate a hydrophobic interior, the hydrophilic region is exposed on the surface of the protein, increasing the proportion of surface charge or hydrophilic groups, and ultimately improving the water solubility of rice protein [35].

Wang et al. [36] prepared a composite of rice protein and whey protein in a 1:1 ratio, which resulted in a composite with a solubility of over 50%. The microstructure of the composite showed that the protein molecules were transformed into dispersed, angular, granular structures. Li et al. [37] utilized pH cycling to prepare a composite of rice protein and walnut protein, which increased the solubility of rice protein to over 80%. The hydrophobic groups of the two proteins were buried and included internally, while the charged groups were exposed externally. The resulting composite had an increased zeta potential and reduced surface hydrophobicity. Manhee et al. [38] found that soy protein isolate can also improve the water solubility of rice protein, increasing its solubility from 25.8% to 68.4%.

In addition, the acylation transfer reaction can also be assisted by transglutaminase to cross-link rice glutelin and another heterologous protein [39]. For example, He et al. [40] promoted the cross-linking reaction between rice glutelin and casein by using transglutaminase, which improved the microstructure of rice glutelin and increased its solubility performance.

### 3.4. Modification and Solubility Enhancement of Rice Protein by Phosphate

Phosphate modification is also a commonly used method to improve the solubility of proteins, and its reaction mechanism is shown in Figure 5. The phosphate group can form hydrogen bonds with the hydroxyl groups on serine, threonine, or tyrosine residues, and increases the number of negative charges, thereby enhancing the electrostatic repulsion between protein molecules and reducing the surface tension of the emulsion, ultimately improving the dispersion of proteins in the emulsion system [41,42]. However, the phosphate modification is currently plagued by the issue of a lengthy reaction time, which can easily cause irreversible aggregation and denaturation of proteins [43]. Therefore, the wet-heat method or microwave method is often used to assist the phosphorylation modification of rice protein to increase the efficiency of the phosphorylation reaction.

Wet-heat-assisted phosphorylation modification involves mixing rice protein with a phosphate solution and incubating it in a 55 °C water bath. This method is effective, simple to operate, and has certain industrial prospects [44]. Hu et al. [45] conducted a wet-heat-assisted phosphorylation reaction on 2% rice bran protein (reaction conditions were 0.1 mol/L sodium trimetaphosphate, pH 9.0, and 55 °C). The conformation of rice bran protein changed, the maximum fluorescence emission wavelength (λ_max_) shifted to blue, and the surface hydrophobicity and fluorescence intensity were significantly increased. Meanwhile, sodium trimetaphosphate introduced more negative charges on the surface of rice bran protein molecules, increasing their hydrophilicity and the repulsion between protein molecules, thereby improving the solubility of rice bran protein.

Microwave-assisted heating can also improve the efficiency of a phosphorylation reaction. Utilizing the enormous energy generated by electromagnetic waves accelerates the unfolding of protein molecules and the exposure of phosphorylation sites, thereby promoting the interaction between phosphate groups and protein side chains [46,47]. Hadidi et al. [48] used microwave-assisted phosphorylation (at a power of 590 W for 155 s) to increase the degree of protein phosphorylation, shorten the reaction time, and improve the solubility and other functional properties of the protein.

### 3.5. Modification and Solubility Enhancement of Rice Protein by Deamidation Method

The modification of proteins through deamidation is typically carried out under acidic or alkaline conditions, or enzymatic catalysis, where the amide groups on the side chains of basic amino acid residues (such as asparagine or glutamine residues) are cleaved to form carboxylic acid groups (thus becoming aspartic acid residues and glutamic acid residues), as shown in Figure 6. Deamidation of proteins can cause protein unfolding or conformational rearrangement while reducing the formation of hydrogen bonds within or between protein subunits caused by the amide groups, which in turn reduces the aggregation between molecules or subunits and enhances the hydrogen bond or hydrophilic interaction between proteins and water molecules [49]. Additionally, deamidation increases the electrostatic repulsion between protein chains and decreases the surface hydrophobicity of the subunits [50].

Guan et al. [51] used the deamidation method under alkaline conditions to obtain highly soluble rice bran protein, by conducting a 30 min deamidation reaction at pH 8.0 and 100 °C. Another approach for deamidation modification is acid-catalyzed deamidation, which is often characterized by the mild properties of organic acids and can prevent issues such as the breakdown of peptide bonds and the isomerization of amino acid residues brought on by excessive deamidation [52]. Li et al. [53] adjusted the pH of glutelin solutions to 4.0 using malic acid and citric acid, and then subjected them to deamidation treatment, resulting in an increase in solubility of glutelin from 7.79% to 39.13% and 26.06%, respectively.

In addition, enzyme-assisted protein deamidation has high reaction specificity and food safety. This method can increase the surface electrostatic charge of the protein, reduce the hydrogen bonds within the molecule, form hydrophilic carboxyl groups, and promote protein unfolding [54]. Liu et al. [49] found that deamidation of rice glutelin by glutamine transamidase could increase its solubility in neutral or slightly acidic solutions (i.e., pH 5–7). Chen et al. [55] discovered that glutamine transamidase could prevent excessive hydrolysis caused by chemical or other protease treatments while increasing the water solubility of glutelin.

### 3.6. Modification and Solubility Enhancement of Rice Protein by Enzymatic Hydrolysis Method

The enzymatic hydrolysis method can break down protein into short-chain peptides or amino acids, and reduce intra- or inter-molecular cross-linking, particularly the formation of inter-subunit disulfide bonds. The small molecules generated by hydrolysis not only exhibit good water solubility, but also possess biological activities such as anti-cancer, anti-hypertension, and immune regulation [56]. Table 1 shows commonly used enzymes, reaction conditions, and modification effects for modified rice protein. Among these proteases, alkaline protease has the highest degree of hydrolysis (DH), and the modification and solubilization of rice protein are the most significant, with solubility as high as 94%. This is due to the fact that alkaline protease has more enzyme hydrolysis sites, resulting in a large number of hydrolysis of hydrophobic amino acid residues, thus improving the solubility of rice protein [57].

The advantages of enzyme hydrolysis include mild reaction conditions, easy control, and high specificity. However, traditional enzymatic hydrolysis can result in bitter peptides (mainly composed of peptides containing Lys, Leu, and Val residues), reduce the emulsifying ability of protein, and has a low hydrolysis efficiency and extended reaction time [58,59]. Physical methods such as ultrasound, high pressure, or the construction of a dual-enzyme system can be used to not only speed up the enzymatic reaction time and increase the efficiency of enzymatic hydrolysis but also to reduce bitter peptides and increase the emulsification and emulsifying stability of the protein.

Yang et al. [60] show that ultrasound-assisted enzymatic hydrolysis can significantly improve the efficiency of rice protein processing. The cavitation effect generated by ultrasound can promote the stretching of enzymatic protein molecules, reduce disulfide bond content and the hydrophobicity of the protein surface, increase the dissolution rate of soluble protein particles or protein molecules, and enhance the contact frequency between protein and enzyme [61], thus improving the enzymatic hydrolysis efficiency. Chang et al. [62] found that ultrasound-assisted enzymatic hydrolysis can promote the exposure of cleavage sites for papain, reduce the hydrolysis time for 1 h, and the enzyme dosage amount by 1.4 times. At the same time, ultrasound-assisted enzymatic hydrolysis increased the water solubility of rice bran protein by nearly 2 times, while the emulsifying stability decreased by 38.25%. This is because the low molecular weight peptides formed by long-time enzymatic hydrolysis (3 h) could not stabilize the oil–water interface, which in turn has an adverse effect on protein emulsification.

High-pressure assistance can improve the solubility and emulsifying stability of rice protein by enhancing enzymatic hydrolysis. Liu et al. [63] found that a pressure of 300 MPa can increase the hydrolysis of rice protein by alkaline protease, improve the solubility of rice protein by nearly 1.7 times, and increase the emulsifying property and emulsifying stability by 2 times and 3 times, respectively. This is due to the changes in the tertiary and quaternary structure of protein molecules caused by ultra-high pressure, which releases smaller soluble proteins and promotes the modification of protease. At the same time, ultra-high pressure shortens the enzymatic hydrolysis time to 15 min and avoids the decrease in protein emulsifying ability. Zhang et al. [64] also found that high-pressure micro fluidization can promote the exposure of hydrophobic groups within the rice bran separation protein, increasing the surface hydrophobicity and molecular diffusion rate, which significantly improves the hydrolysis of rice bran separation protein by neutral protease.

The hydrophobic amino acids in rice protein participate in peptide bond formation at the -amino or carboxyl end during enzymatic hydrolysis, giving the hydrolysate an unfavorable bitter taste [65]. Yan et al. [66] constructed a dual-enzyme system of aminopeptidase and pancreatin, which reduced the bitterness of rice protein hydrolysates. Pooja et al. [67] found that the hydrolysate of rice bran protein pretreated with high hydrostatic pressure can also effectively address issues such as prolonged hydrolysis time and bitter peptides.

**Table 1 molecules-28-04078-t001:** Common enzymes, reaction conditions, and modification effect of modified rice protein.

Enzyme	Condition	Targets	DH	Effect	References
Papain	Enzyme: substrate = 3:100; 4 h; pH = 7.0; 50 °C	Carboxyl terminus of arginine, lysine, and glycine residues	15–32%	Increased the solubility (about 45–94%)	[68]
Trypsin	Enzyme: substrate = 0.89:1000; 2.4 h; pH = 7.6; 52.8 °C	Carboxyl terminal of arginine and lysine residues	8.96%	Increased the solubility (above 75%)	[65]
Alkaline protease	Enzyme: substrate = 1:100; 5 h; pH = 8.0; 65 °C	Carboxyl of hydrophobic amino acid–amide bond of aromatic amino acids	23.8%	Increased the solubility (to 94.78%)	[69,70]
Glutaminase	Enzyme: substrate = 1:250; 12 h; pH = 8.0; 50 °C	Acyl transfer reaction between lysine residue and glutamine residue	4–6%	Increased the solubility (to 78.14%)	[49,71]

## 4. Applications of Modified Rice Protein in Food Processing

Modified rice protein exhibits desirable functional properties such as high solubility, emulsifying ability, gelling ability, and antioxidant activity, which make it a promising ingredient for a wide range of food applications. Modified rice protein can replace allergenic bovine milk protein in dairy products, increasing the potential for developing plant-based dairy products or infant formula [72]. In meat products, it can balance the nutritional value, reduce economic costs, and improve product stability [73]. In baked goods, it can be used to develop gluten-free baked goods, providing more options for people with gluten allergies [74].

### 4.1. Application of Modified Rice Protein in Dairy Product Processing

Modified rice protein retains its low allergenicity, and its emulsifying and encapsulating properties enable the inclusion of vitamins, minerals, probiotics, etc., making it useful in developing infant formula, probiotic fermented milk, cheese, and other dairy products [75].

Modified rice protein can be used in fermented milk products, taking advantage of its good emulsifying and encapsulating properties [76]. Vaniski et al. [77] found that the encapsulation efficiency of thermophilic streptococci by rice bran protein–maltodextrin covalent complexes can reach 90.26%, and the survival rate of thermophilic streptococci in simulated gastric and intestinal fluids is relatively high. Zhang et al. [78] prepared rice protein–pectin composite microcapsules, which exhibited excellent antibacterial activity and could inhibit key enzymes in the tricarboxylic acid cycle and hexose monophosphate pathway of Escherichia coli. After encapsulating probiotics with modified rice protein, it can be directly added as an ingredient to fermented dairy products, and protect the activity of probiotics during food processing and digestion in the human gastrointestinal tract.

Rice protein has low allergenicity and is often used as a substitute for cow’s milk protein in formula powders for lactose intolerant individuals, with certain digestive tolerance, safety, acceptability, and palatability [79]. Amagliani et al. [80] added low-molecular-weight surfactants to hydrolyzed rice protein, mixed it with oils, carbohydrates, and maltodextrins, and developed a rice protein infant formula emulsion formula. The formula reduced the size of fat globules and had high emulsion and thermal stability.

Modified rice protein can also be applied in cheese products, mainly as an active filler embedded in the cheese protein matrix to produce high-quality low-fat cheese [81]. Paximada et al. [82] prepared a water-in-oil (W/O) emulsion of modified rice protein with fat and then homogenized the emulsion with milk to form a water-in-oil-in-water (W/O/W) double emulsion. The process flow diagram of preparing cheese using the W/O/W double emulsion is shown in Figure 7. The double emulsion has a high protein encapsulation efficiency, which can reduce the loss of fat in cheese products and decrease the hardness, thus having a broad application prospect for developing low-fat dairy products. In addition to forming stable water-in-oil emulsions with lipids, the improved functional properties of modified rice protein also facilitate protein gel formation during fermentation, providing possibilities for the development of fermented plant-based cheese [72].

### 4.2. Application of Modified Rice Protein in Meat Products Processing

The processing of meat products is often accompanied by the loss of intracellular and extracellular juice in muscle tissue, as well as a decrease in the water-holding capacity of muscle proteins. Excessive intake of animal protein can also lead to cardiovascular diseases [83] and kidney problems [84]. Modified rice protein has good water-holding properties, which can reduce the loss of fluid in muscle tissue and balance the nutritional value of animal protein, thereby reducing the negative effects of consuming excessive saturated fatty acids on the body [85]. At present, modified rice protein is commonly used in block meat, minced meat, and plant-based meat products.

In the processing of block meat products, modified rice protein can reduce the dehydration shrinkage of whole or large pieces of meat, maintain the integrity of the meat muscle tissue, and improve the product yield. Aqsa et al. [86] mixed chicken chunks with modified rice protein isolate by kneading. The results showed that modified rice protein isolate can increase the protein content of chicken chunks (from 34.99% to 48.49%) and reduce cooking loss (from 12.44% to 3.85%). Zhou et al. [69] found that modified rice protein has a strong antioxidant ability and can effectively reduce lipid oxidation in meat during storage.

Modified rice protein can reduce the loss of fat and moisture in minced meat products and improve sensory quality [87]. Li et al. [88] found that alkaline protease-modified rice protein can form a dense covering film on the surface of the muscle fibers of the sausage. Meanwhile, modified rice protein can form an elastic and hard gel network with meat protein, which can lock fat and moisture in the three-dimensional network structure, reduce the loss of juice, and increase the yield of sausage [89].

Modified rice protein can also be used in the production of plant-based meat products. It has low allergenicity, no beany flavor, and a fatty mouthfeel, and can partially replace soy protein in the production of plant-based meat products [90]. Lee et al. [91] mixed modified rice protein and soy protein in a certain proportion, and used corn starch and wheat flour as additives. They employed a low-moisture extrusion-cooking process with a twin-screw extruder to promote the interaction between proteins, lipids, and carbohydrates, resulting in the production of plant-based meat products with high nutritional value and unique flavor.

### 4.3. Application of Modified Rice Protein in Baked Food Processing

Baked food is typically made from wheat flour through processes such as kneading, fermentation, and baking. However, wheat contains allergenic components such as gluten, which makes it difficult for patients with celiac disease or wheat gluten protein allergies to consume. Therefore, there is a need to develop gluten-free baked food to meet the needs of these special populations. The modified rice protein is free of gluten, has low allergenicity, and has a high water absorption and oil-holding capacity. It can successfully lower the loss of water and oil in baked goods, keep the texture fresh and moist, and raise the amount of protein in them [74].

When mixed with water, sugar, oil, salt, and other ingredients in a certain proportion, modified rice protein can form a sticky and elastic network similar to gluten, which can be baked at high temperatures to make bread, biscuits, or cakes. Sahagún et al. [92] used a recipe containing rice flour, modified rice protein, sugar, milk, pasteurized egg liquid, sunflower oil, and baking powder to make cakes, which improved the texture characteristics of the cakes. Yadav et al. [93] used 5% rice bran concentrate protein instead of refined wheat flour to prepare biscuits, and the results showed that as the concentration of rice bran concentrate protein increased, the physical characteristics of the biscuits, such as diameter, thickness, fracture strength, moisture content, protein content, and ash content, all increased significantly. This may be because rice bran concentrate protein absorbed more water and oil during the baking process, maintaining the moist texture of the biscuits and giving them the best texture in terms of color, taste, and flavor. Honda et al. [94] found that after protease modification, rice protein and gluten protein formed a three-dimensional structure, which increased the volume of gluten-free bread by 22%. Scanning electron microscopy revealed that both the size and quantity of pores in the bread increased, leading to improvements in its rheological and sensory properties [95].

## 5. Conclusions

The modification and solubilization of rice protein are very significant; however, the traditional modification of rice protein is often accompanied by some issues, such as long reaction time, plenty of by-products, insufficient reaction, and so on, while the improved modification method avoids these shortcomings. Modification and solubilization of rice protein by polysaccharides, polyphenols, heterologous proteins, and phosphates are assisted by ultrasound, ultra-high pressure, microwave, enzymes, water bath heating, or alkaline conditions. The modification and solubilization of rice protein by deamidation are assisted by mild organic acids or enzyme-assisted deamidation. However, the control of the degree of deamidation and the safety of protein are still the difficulties of current research. Enzymatic hydrolysis for the modification and solubilization of rice protein is assisted by ultrasound, ultra-high pressure, and dual-enzyme systems. However, it is difficult to realize industrial applications due to the high cost of the experiment. To summarize, under certain physical and chemical conditions, combining rice protein with a certain amount of polysaccharides, polyphenols, heterologous protein, or phosphate anions is the most simple, safe, effective, and low-cost modification method. Among these methods, the polysaccharide reaction can be carried out spontaneously without other chemical reagents, so it has more application potential in improving the water solubility of rice protein.

Modified rice protein has been widely used in the processing of dairy products, meat products, and baked goods due to its good solubility, emulsifying properties, gelling properties, water-holding capacity, and oil-holding capacity. It not only balances nutritional value and meets the protein intake needs of allergic populations, but also improves the deficiencies of animal and plant proteins, increases stability, and extends shelf life. With the in-depth research on the functional properties of modified rice protein, its application scope in food processing is constantly expanding, which can increasingly meet the needs of the food industry and market sectors.

## Figures and Tables

**Figure 1 molecules-28-04078-f001:**
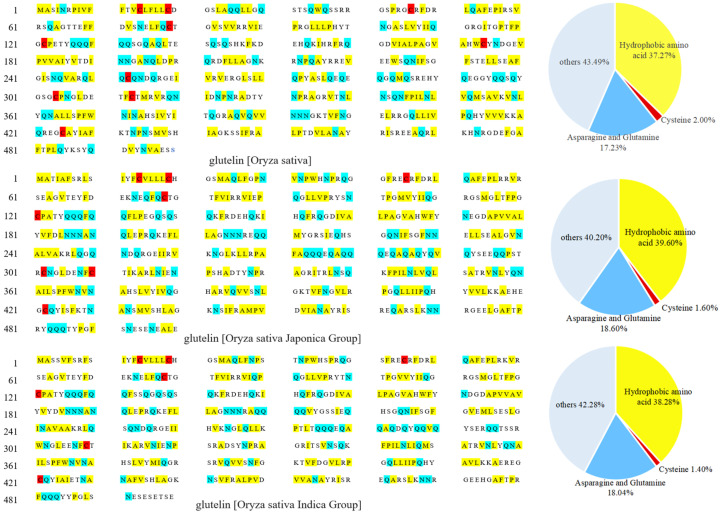
Primary structures and amino acid residue compositions of rice glutelin from three different sources.

**Figure 2 molecules-28-04078-f002:**
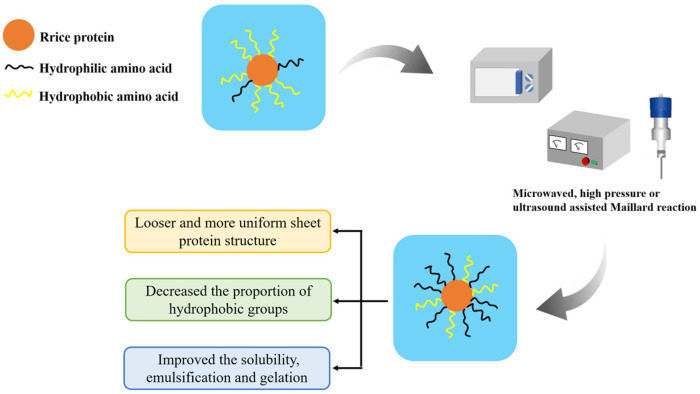
Improvement of solubility of rice protein by using microwaved-, high pressure- or ultrasound-assisted Maillard reaction.

**Figure 3 molecules-28-04078-f003:**
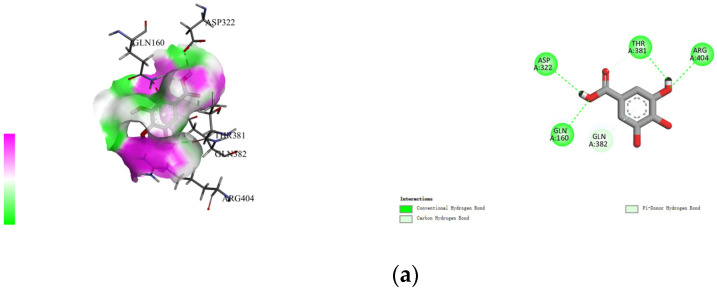
Rice protein–polyphenol molecular docking results. (**a**) gallic acid, (**b**) procyanidins, (**c**) resveratrol, (**d**) ferulic acid.

**Figure 4 molecules-28-04078-f004:**
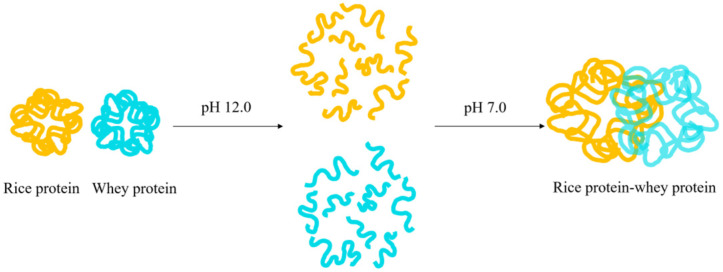
The formation mechanism of the rice protein–whey protein composite.

**Figure 5 molecules-28-04078-f005:**
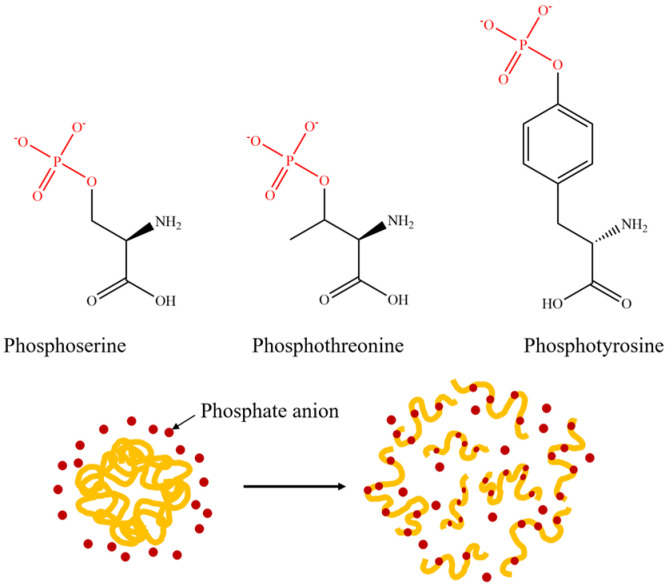
Diagrammatic drawing and structural formula of protein phosphorylation reaction.

**Figure 6 molecules-28-04078-f006:**
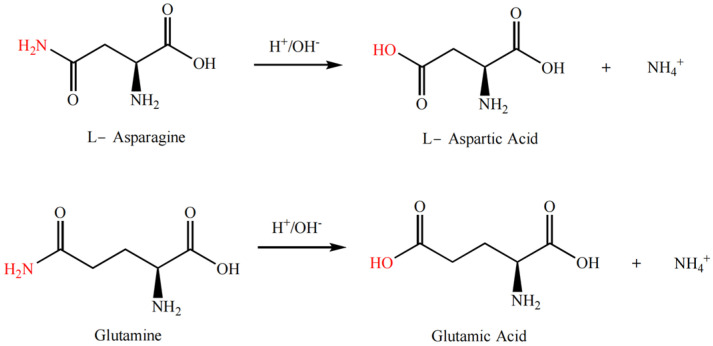
Deamidation reaction formula of rice protein.

**Figure 7 molecules-28-04078-f007:**
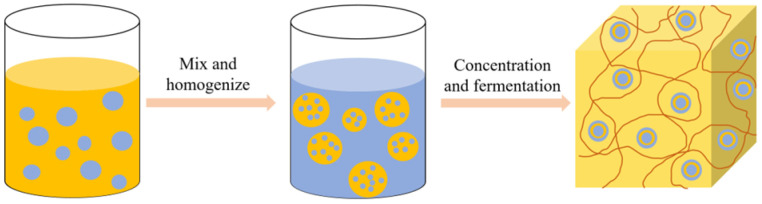
The flow diagram of W/O/W double emulsion preparation of cheese.

## Data Availability

Data are contained within the article.

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
