# Peer review of "Modification and Solubility Enhancement of Rice Protein and Its Application in Food Processing: A Review"

_molecules, 2023, doi:10.3390/molecules28104078_

Round 1

Reviewer 1 Report

the paper titled " Modification and Solubility Enhancement of Rice Protein and Its Application in Food Processing", is a review paper which seem be very interessant and has a value add tp the field. The paper provides an overview of recent methods used for modifying and increasing the solubility of rice protein, and further explores the application of modified rice protein in food production, such as dairy products, meat products, and baked goods

This product has a multuples applications, but the authors must be explain more the innovation and compare among methods of modifcation and which is the best method that is sustainable , econmoc and friendly environment ?

The abstract should be improved to give more information about the motovation to choise this subject.

the conclusion must be improved and should highlight the importance and the contribitio  of the paper in the field.

Add refrences to figures

Author Response

Point 1: This product has a multuples applications, but the authors must be explain more the innovation and compare among methods of modifcation and which is the best method that is sustainable, econmoc and friendly environment?

Response 1: Thanks for your careful checks, we are so sorry for these mistakes. The suggested correction has been made. Please see the first paragraph of the conclusions. The innovation of various modification methods lies in the use of ultrasonic, ultra-high pressure, microwave, and other auxiliary reactions to avoid the shortcomings of traditional modification methods, such as long reaction time, many by-products, insufficient reaction and so on.

“Modification and solubilization of rice protein is very significant, however, the traditional modification of rice protein is often accompanied by some issues, such as long reaction time, plenty by-products, insufficient reaction and so on, while the improved modification method avoids these shortcomings.”

The comparison of various modification methods is found in the fourth and sixth sentences of the conclusion. “However, the control of the degree of deamidation and the safety of protein are still the difficulties of current research.” “However, it is difficult to realize industrial application due to the high cost of the experiment.” “To summary, under certain physical and chemical conditions, combining rice protein with a certain of polysaccharides, polyphenols, heterologous protein, or phosphate anions is the most simple, safe, effective, and low-cost modification method.”

Finally, we summarize the advantages and disadvantages of various modification methods and put forward the best methods which are economical, sustainable, and environmentally friendly. “Among these methods, the polysaccharide reaction can be carried out spontaneously without other chemical reagents, so it has more application potential in improving the water solubility of rice protein.”

Point 2: The abstract should be improved to give more information about the motovation to choise this subject.

Response 2: Thanks for your helpful suggestion, we have added more details. Please see the ABSTRACT. We point out that the motivation for choosing this article is that there are some shortcomings in the traditional modification methods, and the auxiliary traditional modification methods used in this paper have economic and sustainable application prospects.

“it covers the shortcomings of traditional modification methods and the latest compound improvement methods, compares various modification methods, and puts forward the best sustainable, economical, and environment-friendly method.”

Point 3: The conclusion must be improved and should highlight the importance and the contribution of the paper in the field.

Response 3: Thanks for your helpful comments. According to the referee’s comment, the detailed description has been added in the revised manuscript. Please see the conclusion. The contribution of this paper in this field is to summarize the latest progress of rice protein modification and solubilization, and put forward the most promising modification methods.

Point 4: Add refrences to figures

Response 4: Thanks for your helpful comments. The figures used in this article are all original and there is no need to add references.

Reviewer 2 Report

Overall, this review presents comprehensive work related to rice protein. However, some points are required to revise and add more information.

- More significant numerical data must be added, especially, Section 2.1

- Explanation in molecule mechanisms should be addressed.

- Table 1, the effect should added more information such as percentage of increase or decrease in each parameter.

- Section 3.3, more details in effect of different molecular weights of rice protein must be added such as https://www.mdpi.com/2304-8158/12/4/835

Author Response

Point 1: More significant numerical data must be added, especially, Section 2.1

Response 1: Thanks for your rigorous comment, the correction has been made in the revised manuscript. Please see 2.1, 2.2, 2.6.

2.1:” The solubility of the modified rice protein produced by the process was significantly improved to 90.6%.” “which improved the solubility of rice protein (increased from 12.8% to 35.3%),” “Meng et al. [24] found that microwave-assisted heating assisted protein-polysaccharide binding, which could increase the solubility of rice bran protein to 90.97%.”

2.2:” Wang at al.[29] found that under the condition of pH 9.0, the covalent complex of rice bran protein hydrolysate (10mg/mL) and ferulic acid (1.5mg/mL) had strong emulsifying(35.10%) and antioxidant properties(the DPPH· clearance rate is 49.70% and the ABTS+ clearance rate is 89.04%).”

2.6:” Among these proteases, alkaline protease has the highest degree of hydrolysis (DH), and the modification and solubilization of rice protein is the most significant, with solubility as high as 94%.” “reduce the hydrolysis time for 1 hour and the enzyme dosage amount by 1.4 times. At the same time, ultrasound-assisted enzymatic hydrolysis increased the water solubility of rice bran protein by nearly 2 times, while the emulsifying stability decreased by 38.25%.” “improved the solubility of rice protein by nearly 1.7 times, and increased the emulsifying property and emulsifying stability by 2 times and 3 times respectively.”

Point 2: Explanation in molecule mechanisms should be addressed.

Response 2: Thanks for your rigorous comment, the correction has been made in the revised manuscript. Please see 2.6.

“This is due to the fact that alkaline protease has more enzyme hydrolysis sites, resulting in a large number of hydrolysis of hydrophobic amino acid residues, thus improving the solubility of rice protein” “The cavitation effect generated by ultrasound can promote the stretching of enzymatic protein molecules, reduce disulfide bond content and the hydrophobicity of the protein surface,”

Point 3: Table 1, the effect should added more information such as percentage of increase or decrease in each parameter.

Response 3: Thanks for your rigorous comment, we have added more details in Table 1. There are mainly the degree of enzymatic hydrolysis and the percentage increase of each parameter.

Point 4: Section 3.3, more details in effect of different molecular weights of rice protein must be added such as https://www.mdpi.com/2304-8158/12/4/835

Response 4: Thanks for your rigorous comment. Section 3.3 is the application of rice protein in baked goods, which does not involve the problem of molecular weight, but we have added the molecular weight to section 1, is it okay to do more than that?

Reviewer 3 Report

The review is devoted to the methods for increasing solubility of rice proteins, which is necessary to increase their functional properties. Given the scale of the use of rice in the food industry, this issue is certainly of practical and scientific value. My remarks mainly concern chapter 2.6Modification and solubility enhancement of rice protein by enzymatic hydrolysis method”.

1. The information in this chapter is of a general nature. Of the numerical data, it only shows the hydrolysis conditions used by the authors (Table 1). But no useful conclusion follows from these data. It is necessary to provide the degree of hydrolysis of peptide bonds, data on the amount of insoluble polypeptide fragments, etc.

2. Deep hydrolysis of peptide bonds or degradation of protein structure is favorable for solubility, but the functional properties associated with the macromolecular nature of hydrolysates may be lost. The authors need to discuss this issue. This applies not only to chapter 2.6.

3. It is indicated that the use of ultrasound and high-pressure treatments in combination with enzymatic hydrolysis is very promising. It is necessary to give examples of how many times the process is accelerated, what savings of the enzyme are possible, etc. What explains this? It is necessary to give hypotheses about molecular mechanisms.

4. What protease preparations - neutral, acidic or alkaline - are better to use to increase solubility?

5. Please give examples of bitter peptides in rice protein hydrolysates.

6. The text is carelessly written, there are different font sizes

The review is devoted to the methods for increasing solubility of rice proteins, which is necessary to increase their functional properties. Given the scale of the use of rice in the food industry, this issue is certainly of practical and scientific value. My remarks mainly concern chapter 2.6Modification and solubility enhancement of rice protein by enzymatic hydrolysis method”.

1. The information in this chapter is of a general nature. Of the numerical data, it only shows the hydrolysis conditions used by the authors (Table 1). But no useful conclusion follows from these data. It is necessary to provide the degree of hydrolysis of peptide bonds, data on the amount of insoluble polypeptide fragments, etc.

2. Deep hydrolysis of peptide bonds or degradation of protein structure is favorable for solubility, but the functional properties associated with the macromolecular nature of hydrolysates may be lost. The authors need to discuss this issue. This applies not only to chapter 2.6.

3. It is indicated that the use of ultrasound and high-pressure treatments in combination with enzymatic hydrolysis is very promising. It is necessary to give examples of how many times the process is accelerated, what savings of the enzyme are possible, etc. What explains this? It is necessary to give hypotheses about molecular mechanisms.

4. What protease preparations - neutral, acidic or alkaline - are better to use to increase solubility?

5. Please give examples of bitter peptides in rice protein hydrolysates.

6. The text is carelessly written, there are different font sizes

Author Response

Point 1: The information in this chapter is of a general nature. Of the numerical data, it only shows the hydrolysis conditions used by the authors (Table 1). But no useful conclusion follows from these data. It is necessary to provide the degree of hydrolysis of peptide bonds, data on the amount of insoluble polypeptide fragments, etc

Response 1: Thanks for your rigorous comment, we have added more details in Table 1. There are mainly the degree of enzymatic hydrolysis and the percentage increase of each parameter. The conclusion drawn in Table 1 is added to the penultimate sentence of the first paragraph of 2.6.

“Among these proteases, alkaline protease has the highest degree of hydrolysis (DH), and the modification and solubilization of rice protein is the most significant, with solubility as high as 94%.”

Point 2: Deep hydrolysis of peptide bonds or degradation of protein structure is favorable for solubility, but the functional properties associated with the macromolecular nature of hydrolysates may be lost. The authors need to discuss this issue. This applies not only to chapter 2.6.

Response 2: Thanks for your rigorous comment, we have added more details in the second, third, and fourth paragraph of 2.6.

The second paragraph of 2.6: “but also to reduce bitter peptides and increase the emulsification and emulsifying stability of protein.”

The third paragraph of 2.6: “while the emulsifying stability decreased by 38.25%. This is due to the low molecular weight peptides formed by long time enzymatic hydrolysis (3h) could not stabilize the oil-water interface, which in turn has an adverse effect on protein emulsification.”

The third paragraph of 2.6: “and increased the emulsifying property and emulsifying stability by 2 times and 3 times respectively.” “At the same time, ultra-high pressure shortens the enzymatic hydrolysis time to 15min and avoids the decrease of protein emulsifying ability.”

Point 3: It is indicated that the use of ultrasound and high-pressure treatments in combination with enzymatic hydrolysis is very promising. It is necessary to give examples of how many times the process is accelerated, what savings of the enzyme are possible, etc. What explains this? It is necessary to give hypotheses about molecular mechanisms.

Response 3: Thanks for your rigorous comment, we have added more details in the third paragraph of 2.6.

“reduce the hydrolysis time for 1 hour and the enzyme dosage amount by 1.4 times.”

“This is due to the low molecular weight peptides formed by long time enzymatic hydrolysis (3h) could not stabilize the oil-water interface, which in turn has an adverse effect on protein emulsification.”

Point 4: What protease preparations - neutral, acidic or alkaline - are better to use to increase solubility?

Response 4: Thanks for your rigorous comment, we improved Table 1 by retaining only the proteases that increase the solubility of rice protein. By summarizing Table 1, we concluded that alkaline protein can better improve the solubility and provide an explanation of the molecular mechanism.

Please see the first paragraph of 2.6: “Among these proteases, alkaline protease has the highest degree of hydrolysis (DH), and the modification and solubilization of rice protein is the most significant, with solubility as high as 94%. This is due to the fact that alkaline protease has more enzyme hydrolysis sites, resulting in a large number of hydrolysis of hydrophobic amino acid residues, thus improving the solubility of rice protein”

Point 5: Please give examples of bitter peptides in rice protein hydrolysates.

Response 5: Thanks for your rigorous comment, we have added examples of bitter peptides in the second paragraph of 2.6. “(mainly composed of peptides containing Lys, Leu and Val residues),”

Point 6: The text is carelessly written, there are different font sizes

Response 6: Thanks for your rigorous comment, we examined the article carefully, changed the handwriting to the same size, and adjusted a small mistake. Please see the green mark in the article.